# First and Second-Line Anti-Tuberculosis Drug-Resistance Patterns in Pulmonary Tuberculosis Patients in Zambia

**DOI:** 10.3390/antibiotics12010166

**Published:** 2023-01-12

**Authors:** Ngula Monde, Musso Munyeme, Gershom Chongwe, Jonas Johansson Wensman, Mildred Zulu, Seter Siziya, Rabecca Tembo, Kabengele K. Siame, Obi Shambaba, Sydney Malama

**Affiliations:** 1Department of Biomedical Sciences, Tropical Diseases Research Center, Ndola P.O. Box 71769, Zambia; 2Department of Disease Control, School of Veterinary Medicine, University of Zambia, Lusaka P.O. Box 32379, Zambia; 3Department of Clinical Sciences, Swedish University of Agricultural Sciences, 750 07 Uppsala, Sweden; 4Department of Disease Control and Epidemiology, National Veterinary Institute, 751 89 Uppsala, Sweden; 5Department of Pathology and Microbiology, School of Medicine, University of Zambia, Lusaka P.O. Box 50110, Zambia; 6Department of Public Health, Michael Chilufya Sata School of Medicine, Copperbelt University, Ndola P.O. Box 71191, Zambia; 7Department of Biological Sciences, School of Natural Sciences, University of Zambia, Lusaka P.O. Box 32379, Zambia

**Keywords:** drug resistance, extensively drug-resistant, multidrug, tuberculosis, second-line drugs

## Abstract

Background: Drug-resistant tuberculosis has continued to be a serious global health threat defined by complexity as well as higher morbidity and mortality wherever it occurs, Zambia included. However, the paucity of information on drug-susceptibility patterns of both first-line and second-line anti-tuberculosis (anti-TB) drugs, including the new and repurposed drugs used in the management of drug-resistant tuberculosis in Zambia, was the major thrust for conducting this study. Methods: A total of 132 bacteriologically confirmed TB isolates were collected from patients with pulmonary TB during the period from April 2020 to December 2021 in Southern and Eastern Provinces of Zambia. Drug-resistance profiles were determined according to four first-line and five second-line anti-TB drugs. Standard mycobacteriological methods were used to isolate and determine phenotypic drug susceptibility. Data on the participants’ social–demographic characteristics were obtained using a pre-test checklist. Results: Overall, the prevalence of resistance to one or more anti-TB drugs was 23.5% (31/132, 95% CI: 16.5–31.6%). A total of 9.8% (13/132, 95% CI: 5.3–16.2%) of the patients had multidrug-resistant TB and 1.2% were new cases, while 25.5% had a history of being previously treated for TB. Among those with mono-resistant TB strains, isoniazid (INH) resistance was the highest at 9.8% (13/132, 95% CI: 5.3–16.2%). Two (2/31) (6.5%) XDR-TB and one (1/31) (3.2%) pre-XDR-TB cases were identified among the MDR-TB patients. Previously treated patients were 40 times more likely (OR; 40.3, 95% CI: 11.1–146.5%) to have drug-resistant TB than those who had no history of being treated for TB. Conclusion: This study has established a high rate of multidrug-resistant TB and has further identified both pre-XDR- and XDR-TB. There is a need to intensify surveillance of MDR- and XDR-TB to inform future guidelines for effective treatment and monitoring.

## 1. Introduction

Drug-resistant tuberculosis (DR-TB), in particular rifampicin (RIF)-resistant TB (RR-TB), multidrug-resistant (MDR) TB and extensively drug-resistant (XDR) TB continue to be a major public health threat in several countries and threaten global TB control and prevention [1]. TB drug resistance is classified into five categories, namely: isoniazid-resistant TB, RR-TB, MDR-TB, pre-XDR-TB and XDR-TB. MDR-TB is TB that is resistant to rifampicin (RIF) and isoniazid (INH), the two most effective first-line anti-TB drugs. Pre-XDR-TB is MDR-TB that is resistant to any fluoroquinolone, whereas extensively drug-resistant (XDR) TB is defined as MDR-TB plus resistance to any fluoroquinolone and at least one additional group A drug (bedaquiline and linezolid) [2].

Countries that have witnessed the emergence of multidrug-resistant tuberculosis (MDR-TB) and extensively drug-resistant tuberculosis (XDR-TB) TB strains have reported more serious disease, prolonged infectiousness and a poor prognosis among other complexities [3,4,5,6]. The emergence of MDR-TB and XDR-TB in Zambia poses serious challenges to the control of TB and its prevalence. The highest morbidity and mortality rates of TB are mainly reported in low- and middle-income countries, Zambia included [7].

Incomplete and insufficient treatment regimens can lead to antimicrobial resistance. Earlier detection requires access to care and rapid diagnostic tools, which are limited in many provinces of Zambia. Once MDR-TB treatment is started, adherence and tolerability may be a challenge. Recent evidence suggests that MDR-TB is an important contribution to post-TB lung disease (PTLD), which is responsible for disability and suffering that often require rehabilitation [8,9,10,11,12,13,14].

Globally, 3.4 million people were bacteriologically diagnosed with TB and of these, 2.4 million (71%) were tested for rifampicin resistance. Among those tested, 141,953 had MDR-TB/RR-TB and 25,038 had pre-XDR- or XDR-TB. Only one in three people with MDR-TB/RR-TB were enrolled in treatment in 2021. The treatment success rate for MDR-TB/RR-TB was 60% and among the WHO regions, it ranged from 57% in Europe to 72% in the Eastern Mediterranean [15].

Early diagnosis and prompt treatment of TB, including DR-TB, is critical in control of all forms of TB [16,17]. Universal access to drug-susceptibility testing (DST) for DR-TB patients is one of the WHO strategies to reduce the disease burden and start patients on appropriate treatments.

In Zambia, the diagnosis of DR-TB is currently being carried out using GeneXpert MTB/RIF (Xpert MTB/RIF) testing, line probe assay (LPA) (first- and second-line) and phenotypic culture-based drug-susceptibility testing. Primarily, all samples with rifampicin resistance detected on GeneXpert machines are referred to culture facilities for confirmation of MDR-TB as well as the determination of resistance to fluoroquinolones. The National Tuberculosis and Leprosy Control Programme has adopted the new WHO guidelines for treatment of DR-TB, which includes newer drugs such as bedaquiline (BDQ), linezolid (LZD), clofazimine (CFZ) and delamanid (DLM). The current DST method being used for second-line drugs in Zambia (LPA MTBDRsl) does not detect resistance to these newer drugs. Therefore, current DR-TB treatment is not guided by laboratory-confirmed DST results, resulting in patients receiving sub-optimal treatment. Resistance to the new and repurposed TB drugs is now emerging and has been reported elsewhere [18,19,20,21,22]. Lack of DST in these second-line drugs leads to the potential for unidentified drug resistance to the new drugs. Phenotypic and genotypic drug-susceptibility testing to anti-TB drugs guides DR-TB case management and greatly reduces the risk of developing resistance and increases the effectiveness of the treatments. Therefore, the present study aimed to assess first-line and second-line anti-TB drug-resistance patterns in pulmonary TB patients in Zambia in order to guide patient treatment and provide monitoring options.

## 2. Results

### 2.1. Social-Demographic Characteristics of Study Participants

A total of 132 bacteriologically confirmed TB patients were enrolled in the study. The majority, 71.2% (94/132), of the patients were males and 65.9% (87/132) were in the age group of 25-to-44 years with a mean age of 39.15 (SD = 12.34) years. Forty-four percent (59/132) were co-infected with HIV. The majority of the patients came from the Eastern Province, 59.1% (78/132), followed by the Southern Province, 40.9% (54/132) (Table 1).

### 2.2. Drug-Resistance Pattern to First-Line Anti-TB Drugs among TB Patients

The prevalence of drug resistance to first-line drugs among TB patients was 23.5% (31/132, 95% CI: 16.5–31.6%). A total of 9.8% (13/132, 95% CI: 5.3–16.2%) of the patients had multidrug-resistant TB, followed by isoniazid resistance 9.8% (13/132, 95% CI: 5.3–16.2%), rifampicin mono-resistance 3.0% (4/132, 95% CI: 0.8–7.6%) and the least being streptomycin resistance 0.8% (1/132, 95% CI: 0.001–4.1%) (Appendix A). A total of 76.5% (101/132, 95% CI: 68.4–83.5%) of the cases were susceptible to all four first-line drugs (Table 2).

### 2.3. Drug Resistance to Second-Line Anti-TB Drugs among First-Line Drug-Resistant Patients

Resistance to any second-line anti-TB drugs was identified in 35.5% (11/31, 95% CI: 19.2–54.6%) of the DR-TB patients. Resistance to both bedaquiline and clofazimine was observed in 12.9% (4/31, 95% CI: 3.6–29.8%) of the patients. Resistance to clofazimine only was also observed in 12.9% (4/31, 95% CI: 3.6–29.8%) of patients while resistance to bedaquiline, clofazimine, levofloxacin and moxifloxacin was observed in 6.5% (2/31, 95% CI: 0.8–21.4%) of the patients. Resistance to moxifloxacin only (1.0 µg/mL and 0.25 µg/mL) was observed in 3.2% (1/31, (95% CI: 0.08–16.7%) of the patients (Appendix A). One (3.2%) pre-XDR-TB and two (6.5%) XDR-TB cases were identified among the DR-TB patients. Table 3 summarizes the drug-resistance patterns to second-line anti-TB drugs.

### 2.4. Drug-Resistant Tuberculosis among New and Previously Treated Cases

Out of 9.8% (13/132) MDR-TB cases, 1.2% (1/85) were new cases and 25.5% (12/47) had a history of being previously treated for TB. All 27.7% (13/47) of the isoniazid mono-resistant ((9.8%, (13/132)) had a history of previous treatment. Resistance to all second-line drugs was only seen in previously treated cases. Table 4 and Table 5 summarize the findings.

### 2.5. Factors Associated with Drug-Resistant TB

Patients previously treated for TB were more likely (OR; 40.3, 95% CI: 11.1–146.5%) to have drug-resistant TB than those who had no history of being treated for TB (newly treated patients). Other factors such as sex (OR; 1.22, 95% CI: 0.49–3.02%), age (OR; 2.43, 1.37, 95% CI: 0.46–13.25%, 0.55–3.45%) and HIV status (OR; 1.16, 95% CI: 0.51–2.61%) were not significantly associated with having drug-resistant TB (Table 6).

## 3. Discussion

To our knowledge, this is the first study in Zambia to describe drug-resistance patterns in new and repurposed TB drugs. This was based on the primary aim of the study, which was to assess drug-resistance patterns to first- and second-line anti-TB drugs and the factors contributing to DR-TB in Zambia.

Based on our present findings, the overall prevalence of resistance to one or more first-line anti-TB drugs was 23.5%. This is congruent with earlier studies conducted in Kenya, Ethiopia and China, where the DR-TB prevalence was 26.2%, 23% and 31.1%, respectively [23,24,25]. These similarities in prevalence with our study could be attributed to the high TB burdens and HIV/TB co-infections across all four countries. The prevalence in our study is also relatively higher than those reported by studies conducted in Ethiopia at 11.6% and 5.6% [26,27]. Studies conducted by Ibeh et al. and Saifullah et al. reported a much higher prevalence of 41.1% and 65.9%, respectively [28,29]. We also found that 9.8% of the TB patients had multidrug-resistant TB and 1.2% were new cases, while 25.5% had a history of being previously treated for TB. This is similar to what was reported elsewhere, in which the MDR-TB prevalence was 10.6%, 11.6% and 11.5% [23,27,30]. The MDR-TB prevalence in our study is much higher than those reported by other authors in Ethiopia (3.3%), Kenya (4.8%), China (5.7%), Vietnam (6.9%) and Nigeria (7.7%) [24,27,28,31,32]. On the other hand, other studies have reported a higher prevalence than the findings in our study [26,33,34,35,36]. These variations in TB drug resistance among the different studies in different countries could be attributed to HIV/TB confection, irregular supplies of drugs, the case management of DR-TB patients, the diversity in diagnostic methods employed, poor treatment compliance among patients, sample sizes (small sample sizes lead to an overestimation of proportions) and geographical changes [24,37,38,39].

Among the mono-resistant TB strains, the present study showed that isoniazid (INH) resistance was the highest at 9.8%. This finding is comparable to a prospective cohort study conducted in southern Mexico on pulmonary TB patients, where 9.75% of the patients harbored isoniazid mono-resistant strains [40]. Similarly, Villegas and colleagues reported a prevalence of 8% of isoniazid mono-resistance in Peru [41]. Additionally, Dean et al. reported a prevalence that ranged from 10.7% to 27.2% from aggregated DR-TB data reported to WHO from routine surveillance and periodic surveys from the period from 2003 to 2017 and found that Isoniazid is the most critical drug used in the treatment of active TB, and also used for the preventive treatment of TB [42]. Isoniazid mono-resistance is the most common form of DR-TB in the world [39,43,44]. INH mono-resistance escalates the chances of poor treatment outcomes and a progression to MDR-TB [45,46,47]. Most laboratories in Zambia lack the appropriate tools to diagnose INH resistance since the most commonly used GeneXpert MTB/RIF assay detects only RIF resistance. Line probe assay (LPA), a rapid molecular diagnostic test endorsed by WHO in 2008, can detect both INH and RIF resistance, but it is only available at the three TB reference laboratories in Zambia. Therefore, this contributes to the increase in the rate of INH mono-resistant TB cases, as a majority of the cases are misdiagnosed and, subsequently, mismanaged. Isoniazid mono-resistance is associated with higher treatment failures and relapse rates, and the strains may evolve into drug-resistant strains. Therefore, detection of INH resistance with appropriate therapy can significantly increase treatment outcomes in INH-resistant populations and reduce progression to MDR-TB [45,47,48].

More worrying is that our study has exhibited a high proportion of resistance to second-line anti-TB drugs among drug-resistant patients (35.5%). On the contrary, a study conducted by Dagne et al. reported a much lower proportion of resistance to second-line anti-TB drugs of 5.6% [26]. About 12% of the MDR-TB patients in our study had resistance to both bedaquiline and clofazimine. Several studies by other authors have reported cross-resistance to bedaquiline and clofazimine [19,49,50,51,52,53,54]. A phenotypic and genotypic study of drug-resistant *Mycobacterium tuberculosis (mtb)* isolates from cohort studies in KwaZulu-Natal, South Africa, also reported bedaquiline and clofazimine cross-resistance in MDR-TB patients and on the extent of the cross-resistance [21]. Bedaquiline (BDQ) and clofazimine (CFQ) are the most vital second-line anti-TB drugs in the treatment of DR-TB and are widely used in Sub-Saharan Africa, including Zambia. However, DST is not routinely performed [21,49]. BDQ and CFZ share the same efflux pump system (bacterial membrane proteins) MmpS5 and MmpL5, and exposure to CFZ may increase efflux-based resistance, resulting in cross-resistance between the drugs [50,52,55,56].

Another important aspect worth noting in our study is that we identified two (2; 6.5%) XDR-TB cases and one (1; 3.2%) pre-XDR-TB case among the MDR-TB patients. These findings are comparable to a laboratory-based surveillance study conducted in Ethiopia [27], in which the prevalence of pre-XDR-TB was found to be 3% in MDR-TB patients. Another study conducted in China reported an XDR-TB prevalence of 6.8% [57]. The identification of resistance to second-line drugs, particularly XDR-TB, is quite alarming, and this could be attributed to limited DST for second-line drugs in Zambia, as only fluoroquinolones are tested. The current DR-TB patient treatment is not fully guided by laboratory-confirmed DST results; hence, patients are treated empirically. This creates the possibility for unidentified drug resistance to second-line drugs. The use of phenotypic drug-susceptibility testing and molecular tools, such as next-generation sequencing, to guide DR-TB treatments minimizes the risk of the further development of resistance and maximizes the effectiveness of treatments.

The proportions of DR-TB among newly and previously treated patients were 3.5% and 59.6%, respectively. Our study also showed that previously treated patients were 40 times more likely to have drug resistance (OR 40.28; 95% CI: 11.1–146.5%) than those who had never been treated for TB. These findings are in line with results reported by other studies conducted elsewhere [23,29,33,58,59,60,61,62]. The possible reasons for high DR-TB in previously treated cases could be due to poor treatment compliance and/or an inadequate supply of the drugs [63].

The national DR-TB surveys conducted in 2001 and 2008 reported a relatively low prevalence of MDR-TB in Zambia. In 2001, the prevalence of MDR-TB based on the national DR-TB survey was 1.2% in new patients and 1.8% in previously treated TB patients [64]. Similarly, the 2008 national drug-resistance survey showed no prevalence for MDR-TB in new cases and 6.5% in previously treated TB cases. Resistance to any drug was 9.8% among new TB cases [65]. In addition, a recent review of national data collected via a routine surveillance system during a period of 11 years (from 2000 to 2011) showed a fourfold increase in the number of MDR-TB cases diagnosed each year, from 18 in 2000 to 85 in 2011 [66]. Other studies on the prevalence of drug resistance in Zambia include those conducted in 13 Zambian prisons and a cross-sectional explorative study on the prevalence of rifampicin resistance in TB patients at Livingstone Central Hospital [67,68]. The prevalence of MDR-TB in prison inmates was 9.5% [68], and that of rifampicin resistance in the latter study was 5.9% [67]. A retrospective study on DR-TB in the northern region of Zambia found a prevalence of 53% MDR-TB, 32% rifampicin mono-resistance and 1.7% pre-XDR-TB [34]. Another decade-long (2010–2022) retrospective study of TB notifications and mortality trends showed an increase in DR-TB cases, with an average annual rate of 25.0 [69]. Statistics from these previous studies reflect an increase in DR-TB even when compared to the findings in this study.

Although there was no significant association between drug-resistant TB and HIV in our study, evidence exists that shows the link between drug resistance and HIV [70,71,72,73,74]. The surge of MDR-TB occurrence in high HIV settings is critical to public health [75]. TB and HIV co-infection enhances the risk of harboring and acquiring drug-resistant strains, particularly in high TB-burden settings [1]. Adherence to treatment is a major challenge due to higher pill burden, overlapping or additive adverse reactions (ADRS) and drug-to-drug interactions [1,76]. Additionally, some biological mechanisms contribute to drug resistance in HIV-infected individuals. Impaired drug absorption is one of the reasons for the ineffectiveness of TB treatment, particularly with drugs such as rifampicin and ethambutol [74,77]. Therefore, active drug monitoring and management should be an integral part of monitoring DR-TB and HIV co-infected patients.

## 4. Materials and Methods

### 4.1. Study Design

A cross-sectional study was conducted in the Southern and Eastern Provinces of Zambia from April 2020 to December 2021. Study sites comprised the Namwala, Chipata and Lundazi Districts (Figure 1).

### 4.2. Study Population

All bacteriologically confirmed (via Xpert MTB/RIF assay or smear microscopy) TB patients in selected health facilities in Namwala, Chipata and Lundazi were enrolled in the study.

Inclusion Criteria: Voluntary adult (≥15 years) bacteriologically confirmed TB patients (new and previously treated cases) were included.

Exclusion Criteria: Presumptive TB patients with a negative Xpert MTB/RIF assay or smear microscopy results and critically ill patients were excluded.

### 4.3. Data Collection

A pre-test data collection sheet was used to collect social–demographic data of each study participant. Data were collected by health workers in the TB ward at the selected health facilities.

### 4.4. Specimen Collection

A morning sputum sample, approximately 2–10 mL, was collected from each study participant and stored in a sterile sputum container with a tight-fitted lid or cap. The samples were kept at 2 to 8 °C and later triple-packaged and transported using cold-chain protocols to the Tropical Diseases Research Centre (TDRC) TB Regional Reference Laboratory for culture and DST.

### 4.5. Laboratory Analyses

**Sputum sample processing:** Sputum samples were processed using the NaOH-NALC method [78], as previously described [30,79]. Briefly, 4% NaOH was mixed with an equal volume of 2.9% sodium citrate solution, and N-acetyl L-cysteine (NALC) was used as a decontaminant. The sediment was reconstituted to 2 mL with a pH of 6.8 phosphate buffer and inoculated in 7H9 Middlebrook media in the Mycobacterium Growth Indicator Tube (MGIT) system (Becton Dickinson Diagnostic Systems, Sparks, MD, USA) for 42 days. Follow-up methods for positive cultures were employed using blood agar plates (for sterility checks), Ziehl–Neelsen microscopy and Capilia TB Neo tests (TAUNS laboratories. Inc. 761-1, Kamishma, Izunokuni, Shizuoka 410-2325) for differentiation of *Mycobacterium tuberculosis* complex (MTBC) from nontuberculous mycobacteria (NTM).

**Phenotypic First- and Second-Line Drug Susceptibility Testing:** DST was performed following the WHO technical manual for DST of medicines used in the treatment of TB [2]. Susceptibility testing for first-line and second-line agents was performed using the BACTEC MGIT 960 system, which consists of a growth control tube and one tube for each drug, as previously described [80,81]. The critical concentrations for the drugs were as follows for first-line drugs: streptomycin (STR) 1.0 µg/mL; isoniazid (INH) 0.1 µg/mL; rifampicin (RIF) 1.0 µg/mL; and ethambutol (EMB) 5.0 µg/mL. For second-line drugs: moxifloxacin (MOX) 1.0 µg/mL and 0.25 µg/mL; levofloxacin (LEVO) 10 µg/mL; bedaquiline (BDQ) 1.0 µg/mL; linezolid (LZD) 10 µg/mL; and clofazimine (CFZ) 1.0 µg/mL. If the test drug was active against the mycobacteria, it would inhibit growth; while in the control tube (drug-free tube), growth would be uninhibited. Growth was automatically monitored by the MGIT 960 instrument and was recorded using growth units (GU). When the growth control reached a GU of > 400 at 4–13 days for both first- and second-line drugs, the instrument compared the GU of the drug-containing tubes with the growth control tube. Results of either susceptibility or resistance were interpreted based on the GU of the drug-containing tube. A GU of < 100 was interpreted as susceptible (S) while a GU > 100 was interpreted as resistant (R).

### 4.6. Quality Control

Standard operating procedures (SOPs) were followed for culture, identification and DST. *Mycobacterium tuberculosis* H37Rv reference strain (ATTCC 27294), which is susceptible to all anti-TB drugs, was used as a quality control strain and included in each batch as a positive control for all procedures.

### 4.7. Statistical Analysis

The collected data were entered into a Microsoft Excel spreadsheet and exported into SPSS version 21 (IBM Corp, Armonk, NY, USA) statistical software. Categorical variables were presented as frequencies and percentages. For demographic variables, Pearson’s chi-squared test and, where necessary, Fisher’s exact test were used to test for associations between independent variables and the outcome variables. For the risk factors associated with TB, a bivariate analysis was conducted using logistic regression to build a model. Results with a *p*-value less than 0.05 were considered statistically significant.

## 5. Conclusions

This study has shown a high proportion of MDR-TB and INH resistance. It has also highlighted the resistance to second-line drugs with the detection of XDR-TB and pre-XDR-TB among the MDR-TB cases. Furthermore, previously treated patients were more likely to have drug resistance than those who had never been treated. The use of second-line drugs to treat drug-resistant TB is likely to increase in XDR-TB. Therefore, accurate measures of the incidence, prevalence and determinants of XDR-TB are needed to guide the treatment response. The presence of drug resistance in both first-line and second-line drugs implies that timely, feasible and rapid DST is needed to guide patient management and prevent the further progression of resistance. Additionally, a rollout of the new and repurposed second-line drugs should be informed by laboratory-confirmed DSTs.

## Figures and Tables

**Figure 1 antibiotics-12-00166-f001:**
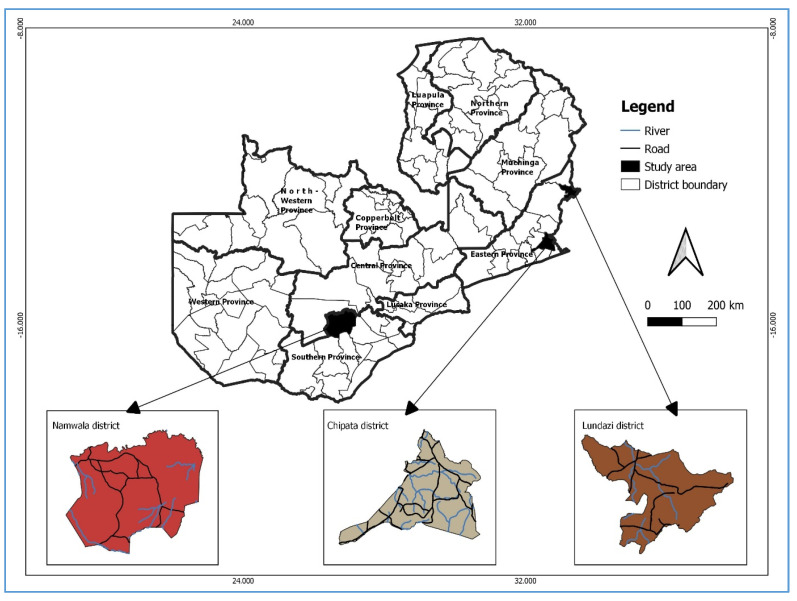
Map of Zambia depicting the three study sites.

**Table 1 antibiotics-12-00166-t001:** Social–demographic characteristics of study participants.

Variable	Category	Frequency (N = 132)	Proportion (%)	95% CI
Gender	Male	94	71.2	62.7–78.8
	Female	38	28.8	21.2–37.3
Age	0–24 years	14	10.6	5.9–17.2
	25–44 years	87	65.9	57.2–73.9
	45+ years	31	23.5	16.5–31.6
HIV Status	Reactive	59	44.7	36.0–53.6
	Non-reactive	73	55.3	46.4–64.0
Address	Urban	85	64.4	55.6–72.7
	Rural	47	35.6	27.5–44.4
Province	Southern	54	40.9	32.4–49.8
	Eastern	78	59.1	50.2–67.6
Reasons for Examination	Diagnosis	87	65.9	57.2–73.9
	Follow up	45	34.1	26.1–42.8
TB Status	Drug Resistant	31	23.5	16.5–31.6
	Drug Susceptible	101	76.5	68.4–83.5

**Table 2 antibiotics-12-00166-t002:** Proportions of DR-TB to first-line anti-TB drugs among TB patients.

First-Line DST	Number (N = 132)	Proportion (%)	95% CI
Multidrug resistant	13	9.8	5.3–16.2
Rifampicin mono- resistant	4	3.0	0.8–7.6
Isoniazid mono- resistant	13	9.8	5.3–16.2
Streptomycin mono- resistant	1	0.8	0.001–4.1
Susceptible (RIF, INH, STR, ETH)	101	76.5	68.4–83.5

DST—drug-susceptibility testing; CI—confidence interval; RIF—rifampicin; INH—isoniazid; STR—streptomycin; ETH—ethambutol; N—number of participants; %—percentage.

**Table 3 antibiotics-12-00166-t003:** Proportions of DR to second-line anti-TB drugs among first-line drug-resistant patients.

Second-Line DST	Number Resistant (N = 31)	Proportion (%)	95% CI
BDQ + CFZ	04	12.9	3.6–29.8
CFZ	04	12.9	3.6–29.8
BDQ + CFZ + LEVO + MOX ^#^	02	6.5	0.8–21.4
MOX *	01	3.2	0.08–16.7
LZD	0	0	-
Susceptible (BDQ, CFZ, LEXO, MOX, LZD)	20	64.5	45.4–80.8

BDQ—bedaquiline, CFZ—clofazimine, LEVO—levofloxacin, MOX—moxifloxacin, LZD—linezolid. * Pre-XDR, N—number of participants, CI—confidence interval, %—percentage, **^#^** XDR-TB.

**Table 4 antibiotics-12-00166-t004:** First-line drug-resistant patterns among new and previously treated cases (N = 132).

DR Pattern	Total (*n* = 132), (%)	New Cases (*n* = 85), (%)	Previously Treated Cases (*n* = 47), (%)
Susceptible	101 (76.5)	82 (96.5)	19 (40.2)
MDR	13 (9.8)	01 (1.2)	12 (25.5)
RR	04 (3.0)	01 (1.2)	03 (6.4)
INH	13 (9.8)	0 (0)	13 (27.7)
STR	01 (0.8)	01 (1.2)	0 (0)

DR—drug resistant; MDR—multidrug resistant; RIF—rifampicin; INH—isoniazid; STR—streptomycin; ETH—ethambutol; *n*—number of participants, %—percentage.

**Table 5 antibiotics-12-00166-t005:** Second-line drug-resistant patterns among new and previously treated cases (N = 31).

DR Pattern	Total (*n* = 31)	New Cases (*n* = 3)	Previously Treated Cases (*n* = 28)
Susceptible	20 (64.5)	03 (100)	17 (60.7)
BDQ + CFZ	04 (12.9)	0 (0)	04 (14.3)
CFZ	04 (12.9)	0 (0)	04 (14.3)
BDQ + CFZ + LEVO + MOX	02 (6.5)	0 (0)	02 (7.1)
MOX	01 (3.2)	0 (0)	01 (3.6)
LZD	0 (0)	0 (0)	0 (0)

DR—drug resistant, BDQ—bedaquiline, CFZ—clofazimine, LEVO—levofloxacin, MOX—moxifloxacin, LZD—linezolid. *n*—number of participants, f—frequency.

**Table 6 antibiotics-12-00166-t006:** Factors associated with drug-resistant TB.

Variable	Level (*n* = 132)	Frequency	DR_TB (%)	OR	95% CI	*p*-Value
Gender	Male	94	23 (24.5)	1.22	0.49–3.02	0.675
	Female	38	8 (21)	1		
Age	0–24 years	14	2 (14.3)	2.43	0.46– 13.25	0.302
	25–44 years	87	20 (23.0)	1.37	0.55–3.45	0.503
	45+ years	31	9 (29.0)	1		
HIV Status	Reactive	59	13 (22.0)	1.16	0.51– 2.61	0.724
	Non-reactive	73	18 (24.7)	1		
Address	Urban	85	16 (18.8)	2.02	0.89– 4.59	0.092
	Rural	47	15 (31.9)	1		
Province	Southern	54	13 (24.1)	0.95	0.42–2.14	0.894
	Eastern	78	18 (23.1)	1		
Treatment History	Previously treated New	47	28 (59.6)	40.3	11.1–146.0	0.001
		85	3 (3.5)	1		

DR—drug resistant, *n*—number of participants; %—percentage; CI—confidence interval, OR—odds ratio.

## Data Availability

Data supporting the results in this study can be made available on request from the corresponding author.

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
