# Peer review of "First and Second-Line Anti-Tuberculosis Drug-Resistance Patterns in Pulmonary Tuberculosis Patients in Zambia"

_antibiotics, 2023, doi:10.3390/antibiotics12010166_

Round 1

Reviewer 1 Report

The manuscript by Monde et al submitted a manuscript entitled “First and second-line anti-tuberculosis drug resistance patterns in Pulmonary Tuberculosis patients in Zambia. Briefly the authors established a high rate of multidrug resistant TB and has further identified both pre-XDR and XDR-TB. There is need to intensify surveillance of MDR and XDR-TB to inform future guidelines for effective treatment and monitoring.

Overall study was conducted very carefully, and introduction, methods, results and discussion were sufficiently described and well presented.The manuscript by Monde et al submitted a manuscript entitled “First and second-line anti-tuberculosis drug resistance patterns in Pulmonary Tuberculosis patients in Zambia. Briefly the authors established a high rate of multidrug resistant TB and has further identified both pre-XDR and XDR-TB. There is need to intensify surveillance of MDR and XDR-TB to inform future guidelines for effective treatment and monitoring.

Overall study was conducted very carefully, and introduction, methods, results and discussion were sufficiently described and well presented.

Author Response

Thank you so much for the review and feedback.

Reviewer 2 Report

Journal  Antibiotics

Manuscript IDantibiotics-2124804

Type Article

Title  First and second-line anti-tuberculosis drug resistance patterns in Pulmonary Tuberculosis patients in Zambia

Authors   Ngula Monde * , Musso Munyeme , Gershom Chongwe , Jonas Johansson Wensman , Mildred Zulu , Seter Siziya , Rabecca Tembo , Kabengele Keith Siame , Obi Shambaba , Sydney Malama

1. “However, the 19 paucity of information on drug susceptibility patterns of anti-tuberculosis (anti-TB) drugs including 20 both first-line and second-line used in the management of drug resistant tuberculosis in Zambia”

Really paucity? Pubmed search of “tuberculosis Zambia resistance” gives 95 results

2. “A total of 132 bacteriologically confirmed TB isolates were collected from patients with 23 pulmonary TB during the period from April, 2020 to December, 2021 in Southern and Eastern provinces of Zambia.”

You wrote in MM that you included all available but this is really strange such a small number of isolates. In any case the collection is small. 1 year 8 months, 2 provinces and only 132 isolates.

Furthermore, collection included both chronic and new cases together. Not sure you can analyse results for such merged collection. Strains from new patients reflect current transmission, while strains from previously treated patients may reflect treatment failure, inadequate treatment, resistance acquired long ago.

3.  Supplement table should be shown for all tested isolates:

1.     Strain ID

2.     Province

3.     Urban/rural

4.     Previously treated or new

5.     Resistance to each tested drug

Author Response

Reviewer 2

  1. “However, the 19 paucity of information on drug susceptibility patterns of anti-tuberculosis (anti-TB) drugs including 20 both first-line and second-line used in the management of drug resistant tuberculosis in Zambia”

Really paucity? Pubmed search of “tuberculosis Zambia resistance” gives 95 results

Response: Most of these studies (95) are on tuberculosis without drug resistance and were not done in Zambia but included co-authors from Zambia and cited data from Zambia. A few studies on drug resistance are based on first line drugs and few second-line drugs (Fuoroquinolones and injectables) using geneXpert , Line probe Assay or phenotypic first-line drug susceptibility testing. Therefore, there is limited data on both firstline and secondline drug susceptibility testing including the WHO recommended new and repurposed drugs used in the management of TB in Zambia.

Revision has also be made to the manuscript.

  1. “A total of 132 bacteriologically confirmed TB isolates were collected from patients with 23 pulmonary TB during the period from April, 2020 to December, 2021 in Southern and Eastern provinces of Zambia.”

You wrote in MM that you included all available but this is really strange such a small number of isolates. In any case the collection is small. 1 year 8 months, 2 provinces and only 132 isolates.

Furthermore, collection included both chronic and new cases together. Not sure you can analyse results for such merged collection. Strains from new patients reflect current transmission, while strains from previously treated patients may reflect treatment failure, inadequate treatment, resistance acquired long ago.

Response 1: We collected samples from two provinces but only one district in southern province and two districts in Eastern province were selected for sample collection hence the small number of isolates.

Response 2: Yes, strains were collected from new TB patients and previously treated TB patients. Most of the patients who were Drug resistant were previously treated patients (Relapses) and were declared cured or completed their most recent firstline TB treatment. Its difficult to tell at this stage whether this category of patients were infected by a new strain or with the old strain since molecular techniques such as sequencing were not employed in this study.

  1. Supplement table should be shown for all tested isolates:
  2. Strain ID
  3. Province
  4. Urban/rural
  5. Previously treated or new
  6. Resistance to each tested drug

Response: The table has been created and will be presented as supplementary materials

Reviewer 3 Report

The authors have done a thorough and critical study of the prevalence of drug resistance to first and second-line TB drugs which are extremely critical to understand the emergence of drug resistance The following are some minor comments at my end for the manuscript:

1. The authors should specify if any of the samples were retrospective or all of them were current (Pre-treatment samples of new patients).

2. For patients with HIV, the authors should further divide the sections and analyze data to understand any observable differences between patients on HIV therapy and Drug resistance to TB. 

3. The authors should include in the discussion section a) A comparison of data from previous years and the highlight of the treatment for HIV has led to any observable differences in the emergence of Drug-resistant Tuberculosis.

Author Response

Reviewer 3

The authors have done a thorough and critical study of the prevalence of drug resistance to first and second-line TB drugs which are extremely critical to understand the emergence of drug resistance The following are some minor comments at my end for the manuscript:

  1. The authors should specify if any of the samples were retrospective or all of them were current (Pre-treatment samples of new patients).

Response: All the samples were currently collected from TB diagnostic facilities.

  1. For patients with HIV, the authors should further divide the sections and analyze data to understand any observable differences between patients on HIV therapy and Drug resistance to TB.

Response: We analysed the data and we could not find any observable differences between patients on HIV therapy and drug resistance. 

  1. The authors should include in the discussion section a) A comparison of data from previous years and the highlight of the treatment for HIV has led to any observable differences in the emergence of Drug-resistant Tuberculosis.

Response: It has been addresses in the discussion section of the manuscript on page 7.